# Face Reconstruction from Voice using Generative Adversarial Networks

**Yandong Wen**
Carnegie Mellon University
Pittsburgh, PA 15213
yandongw@andrew.cmu.edu

**Rita Singh**
Carnegie Mellon University
Pittsburgh, PA 15213
rsingh@cs.cmu.edu

**Bhiksha Raj**
Carnegie Mellon University
Pittsburgh, PA 15213
bhiksha@cs.cmu.edu

## Abstract

Voice profiling aims at inferring various human parameters from their speech, e.g. gender, age, etc. In this paper, we address the challenge posed by a subtask of voice profiling - reconstructing someone's face from their voice. The task is designed to answer the question: given an audio clip spoken by an unseen person, can we picture a face that has as many common elements, or associations as possible with the speaker, in terms of identity?

To address this problem, we propose a simple but effective computational framework based on generative adversarial networks (GANs). The network learns to generate faces from voices by matching the identities of generated faces to those of the speakers, on a training set. We evaluate the performance of the network by leveraging a closely related task - cross-modal matching. The results show that our model is able to generate faces that match several biometric characteristics of the speaker, and results in matching accuracies that are much better than chance. The code is publicly available in https://github.com/cmu-mlsp/reconstructing_faces_from_voices

## 1 Introduction

The challenge of voice profiling is to infer a person's biophysical parameters, such as gender, age, health conditions, etc. from their speech, and a large body of literature exists on the topic [28, 1, 21]. In this paper, we extend this challenge and address the problem: is it possible to go beyond merely predicting a person's physical attributes, and actually reconstruct their entire face from their voice? Effectively a new subtask of voice profiling, the task is designed to answer the question: given an unheard audio clip spoken by an unseen person, can we picture a face image that has as many as possible associations with the speaker in terms of identity?

A person's voice is incontrovertibly statistically related to their facial structure. The relationship is, in fact, multi-faceted. *Direct* relationships include the effect of the underlying skeletal and articulator structure of the face and the tissue covering them, all of which govern the shapes, sizes, and acoustic properties of the vocal tract that produces the voice [22, 38]. Less directly, the same genetic, physical and environmental influences that affect the development of the face also affect the voice. Demographic factors, such as gender, age and ethnicity too influence both voice and face (and can in fact be independently inferred from the voice [1, 15] or the face [17]), providing additional links between the two.

Neurocognitive studies have shown that human perception implicitly recognizes the association of faces to voices [4]. Studies indicate that neuro-cognitive pathways for voices share a common structure with that for faces [7] – the two may follow parallel pathways within a common recognition framework [4, 3]. In empirical studies humans have shown the ability to associate voices of unknown individuals to pictures of their faces [11, 19]. They are seen to show improved ability to memorize

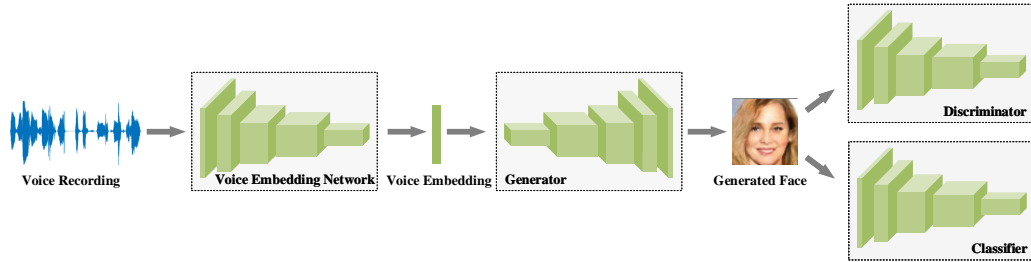

Figure 1: The proposed GANs-based framework for generating faces from voices. It includes 4 major components: voice embedding network, generator, discriminator, and classifier.

and recall voices when previously shown pictures of the speaker's face, but not imposter faces [20, 32, 31].

Given these demonstrable dependencies, it is reasonable to hypothesize that it may also be possible to reconstruct face images from a voice signal algorithmically.

On the other hand, reconstructing the face from voice is a challenging, maybe even impossible task for several reasons [35]. First, it is an ill-posed cross-modal problem : although many face-related factors affect the voice, it may not be possible to entirely disambiguate them from the voice. Even if this were not the case, it is unknown *a priori* exactly what features of the voice encode information about any given facial feature (although one may take guesses [35]). Moreover, the signatures of the different facial characteristics may lie in different spoken sounds; thus, in order to obtain sufficient evidence, the voice recordings must be long enough to have sufficient coverage of sounds to derive all the necessary information. The information containing in a single audio clip may not be sufficient for constructing a face image.

Yet, although *prima facie* the problem seems extremely hard, recent advances in neural network based generative models have shown that they are able to perform similarly challenging generative tasks in a variety of scenarios, when properly structured and trained. In particular, generative adversarial networks (GANs) have demonstrated the ability to learn to generate highly sophisticated imagery, given only signals about the validity of the generated image, rather than detailed supervision of the content of the image itself [23, 30, 40]. We use this ability to learn to generate faces from voices.

For our solution, we propose a simple but effective data-driven framework based on generative adversarial networks (GANs), as illustrated in Fig. 1. The objective of the network is simple: given a voice recording it must generate a face image that plausibly belongs to that voice. The voice recording itself is input to the generator network in the form of a voice embedding vector extracted by a voice embedding network. The generator is trained using a pair of discriminators. The first evaluates if the images it generates are realistic face images. The second discriminator (classifier) verifies that the identity of face image output by the generator does indeed match the actual identity of the speaker.

We present both qualitative and quantitative evaluations of the results produced by our model. The qualitative results show that our framework is able to map the voice manifold to face manifold. We can observe many identity associations between the generated faces and the input voices. The generated faces are generally age and gender appropriate, frequently matching the real face of the speaker. Additionally, given non-speech input the outputs become unrealistic, showing that the learned mapping is at least somewhat specific, in that the face manifold it learns are derived primarily from the voice manifold and not elsewhere. In addition, for different speech segments from the same person, the generated faces exhibit reasonable intra-class variation.

We also propose a number of quantitative evaluation metrics to evaluate the output of our network, based on how specific the model is in mapping voices to faces, how well the high-level attributes of the generated face match that of the speaker, and how well the generated faces match the identity (ID) of the speaker itself. For the last metric (ID matching), we leverage the cross-modal matching task [25], wherein, specifically, we need to match a speech segment to one of the two faces, where one is the true face of the speaker, and another is an "imposter." Our tests reveal that the network is highly specific in generating faces in response to voices, produces quantifiably gender-appropriate faces from voices, and that the matching accuracy is much better than chance, or what may be obtained merely by matching gender. We refer the reader to the experiments section for actual numbers.

Overall, our contributions are summarized as follows:

- We introduce a new task of generating faces from voice in voice profiling. It could be used to explore the relationship between voice and face modalities.

- We propose a simple but effective framework based on generative adversarial networks for this task. Each component in the framework is well motivated.

- We propose to quantitatively evaluate the generated faces by using a cross-modal matching task. Both the qualitative and quantitative results show that our framework is able to generate faces that have identity association with the input voice.

## 2    Related Works

The task of deriving faces from voices relates to several research areas.

**Voice profiling.** There currently exists a significant body of research on deriving personal profile parameters from a person's voice [35]. Many useful characteristics about the speaker can be inferred from their speech, e.g. age [28, 1], emotion [21], identity [6], anthropometric measurement [36], health status [33] etc. These profile parameters may be viewed as providing linkages between a person's voice and their face.

**Face generation using GANs.** There is a long line of research on generating faces using GANs. [29, 12] can be used to generate face images from noise, but it does not consider any conditioning information (like identity) as input. [27, 18] use a discrete label as the condition, but it only works on a closed-set scenario. [8, 2] achieve identity-preserving face generation in an open-set scenario. However, the problem they focus on is within-modal generation problem, where a reference face image of the target identity to be generated is shown to the model. Our task is more challenging, since the conditioning information provided is actually from a different modality, namely voice.

**Voice to face matching.** Cross modal matching between voice and face has become an increasingly popular problem in recent years. It aims at matching a probe input from one modality to a gallery of multiple inputs from the other modality. [25] formulates this task as a $N$-way classification problem. [24, 13, 39] propose to learn common embeddings for the cross-modal inputs, such that the matching can be performed using the learned embeddings. All of these, however, are essentially *selection* problems. Our task requires actual generation is naturally more challenging than cross-modal matching. In part, this is also because many possible generated images could be the expected output for an audio input. There is no unique target output as a supervision signal to train our model.

**Talking face.** Talking face [41, 10] is a recently proposed task. Given a static face image and a speech clip, the goal is to generate a sequence of target face images where the lips are synchronized with the audio. The talking face task is significantly different from our task; in fact the two are actually somewhat orthogonal to each other because talking face focuses on the content of the speech and ignores the identity of the speaker, while our task does the opposite, extracting the identity of speaker and discarding the content of speech.

**Speech to face.** A very recent, yet to be presented publication that has been brought to our notice by the authors also reports on the production of faces from voices [37]. While the problem addressed is similar, the approach claimed is different. Since the codebase and much of the necessary detail remain unpublished at the time of this submission[1], we do not perform comparative assessments.

## 3    The Proposed Framework

Before we begin, we first specify some of the notation we will use. We represent voice recordings by the symbol $v$, using super or subscripts to identify specific recordings. Similarly, we represent face images by the symbol $f$. We will represent the *identity* of a subject who provides voice or face data as $y$. We will represent the true identity of (the subject of) voice recording $v$ as $y^v$ and face $f$ as $y^f$. We represent the function that maps a voice or face recording to its idenitity as $ID()$, *i.e.* $y^v = ID(v)$ and $y^f = ID(f)$. Additional notation will become apparent as we introduce it.

Our objective is to train a model $F(v; \Theta)$ (with parameter $\Theta$) that takes as input a voice recording $v$ and produces, as output a face image $\hat{f} = F(v; \Theta)$ that belongs to the speaker of $v$, *i.e.* such that $ID(\hat{f}) = ID(v)$.

We use the framework shown in Figure 1 for our model, which decomposes $F(v; \Theta)$ into a sequence of two components, $F_e(v; \theta_e)$ and $F_g(e; \theta_g)$. $F_e(v; \theta_e) : v \rightarrow e$ is a voice embedding function with parameter $\theta_e$ that takes in a voice recording $v$ and outputs an embedding vector $e$ that captures all the salient information in $v$. $F_g(e; \theta_g) : e \rightarrow f$ is a *generator* function that takes in an embedding vector and generates a face image $\hat{f}$.

We must learn $\theta_e$ and $\theta_g$ such that $ID(F_g(F_v(v; \theta_e); \theta_g)) = ID(v)$.

## 3.1 Training the network

### 3.1.1 Data

We assume the availability of face and voice data from a set of subjects $\mathcal{Y} = \{y_1, y_2, \cdots, y_k\}$. Correspondingly, we also have a set of voice recordings $\mathcal{V} = \{v_1, v_2, ..., v_N\}$, with identity labels $\mathcal{Y}^v = \{y_1^v, y_2^v, ..., y_N^v\}$ and a set of faces $\mathcal{F} = \{f_1, f_2, ..., f_M\}$ with identity labels $\mathcal{Y}^f = \{y_1^f, y_2^f, ..., y_M^f\}$, such that $y^v \in \mathcal{Y} \; \forall y^v \in \mathcal{Y}^v$ and $y^f \in \mathcal{Y} \; \forall y^f \in \mathcal{Y}^f$. $N$ may not be equal to $M$.

In addition, we define two sets of labels $\mathcal{R} = \{r_1, r_2, ..., r_M \mid \forall i, \; r_i = 1\}$ and $\hat{\mathcal{R}} = \{\hat{r}_1, \hat{r}_2, ..., \hat{r}_N \mid \forall i, \; \hat{r}_i = 0\}$ corresponding $\mathcal{Y}^f$ and $\mathcal{Y}^v$ respectively. $\mathcal{R}$ is a set of labels that indicates that all faces in $\mathcal{F}$ are "real." $\hat{\mathcal{R}}$ is a set of labels that indicates that any faces generated from any $v \in \mathcal{V}$ are synthetic or "fake."

### 3.1.2 GAN framework

In training the model, we impose two requirements. First, the output $\hat{f}$ of the generator in response to any actual voice input $v$ must be a realistic face image. Second, it must belong to the same identity as the voice, *i.e.* $ID(\hat{f}) = f^v$. As explained in Section 1, we will use a GAN framework to train $F_e(.; \theta_e)$ and $F_g(.; \theta_g)$. This will require the definition of *adversary* that provide losses that can be used to learn the model parameters.

We define an adversarial objective. First, the *discriminator* $F_d$ determines if any input image ($f$ or $\hat{f}$) is a genuine picture of a face, or one generated by the generator, *i.e.* assigns any face image ($f$ or $\hat{f}$) to its real/fake label ($r$ or $\hat{r}$). The loss function for $F_d$ is defined as $L_d(F_d(f), r)$ (or $L_d(F_d(\hat{f}), \hat{r})$). Second, *classifier* $F_c$ learns to assign any real face image $f$ to its identity label $y^f$. Accordingly, the loss function for $F_c$ is $L_c(F_c(f), y^f)$. Last, the generator $F_g$ takes in a voice recording $v$ and attempts to generate any face image $\hat{f}$ that can be classified to real label $r$ and identity label $y^v$ by $F_d$ and $F_g$, respectively. The corresponding loss function for $F_g$ is $L_d(F_d(F_g(v)), r) + L_c(F_c(F_g(v)), y^v)$.

In our implementation, we instantiate $F_e(v; \theta_e)$, $F_g(e; \theta_g)$, $F_d(f; \theta_d)$ and $F_c(f; \theta_c)$ as convolutional neural networks, shown in Fig. 1. $F_e$ is the component labeled as *Voice Embedding Network*. $v$ is the Mel-Spectrographic representations of speech signal. The output of the final convolutional layer is pooled over time, leading to a $q$-dimensional vector $e$. $F_g$ is labeled as *Generator*. $f$ and $\hat{f}$ are RGB images with the same resolution of $w \times h$. $F_d$ and $F_c$ are labeled as *Discriminator* and *Classifier*, respectively. The loss functions $L_d$ and $L_c$ of these two components are the cross-entropy loss.

### 3.1.3 Training the network

The training data comprise a set of voice recordings $\mathcal{V}$ and a set of face images $\mathcal{F}$. From the voice recordings in $\mathcal{V}$ we could obtain the corresponding generated face images $\hat{\mathcal{F}} = \{\hat{f} = F_g(F_e(v)) \mid \forall v \in \mathcal{V}\}$.

The framework is trained in an adversarial manner. To simplify, we use a pretrained voice embedding network $F_e(v; \theta_e)$ from a speaker recognition task, and freeze the parameter $\theta_e$ when training our framework. $F_d$ is trained to maximize $\sum_{i=1}^{M} L_d(F_d(f_i), r_i) + \sum_{i=1}^{N} L_d(F_d(\hat{f}_i), \hat{r}_i)$ with fixed $\theta_e$, $\theta_g$ and $\theta_c$. Similarly, the $F_c$ is trained to maximize $\sum_{i=1}^{M} L_c(F_c(f_i), y_i)$ with fixed $\theta_e$, $\theta_g$ and $\theta_d$. The

$F_g$ is trained to maximize $\sum_{i=1}^{N} L_d(F_d(\hat{f}_i), r_i) + L_c(F_c(\hat{f}_i), y_i^v)$ with $\theta_e$, $\theta_d$ and $\theta_c$ fixed, where $\hat{f}_i = F_g(F_e(v_i))$. The training pipeline is summarized in Algorithm 1.

---

**Algorithm 1** The training algorithm of the proposed framework

---

**Input:** A set of voice recordings with identity label $(\mathcal{V}, \mathcal{Y}^v)$. A set of labeled face images with identity label $(\mathcal{F}, \mathcal{Y}^f)$. A voice embedding network $F_e(v; \theta_e)$ trained on $\mathcal{V}$ with speaker recognition task. $\theta_e$ is fixed during the training. Randomly initialized $\theta_g$, $\theta_d$, and $\theta_c$

**Output:** The parameters $\theta_g$.

1: **while** not converge **do**
2:     Randomly sample a minibatch of $n$ voice recordings $\{v_1, v_2, ..., v_n\}$ from $\mathcal{V}$
3:     Randomly sample a minibatch of $m$ face images $\{f_1, f_2, ..., f_m\}$ from $\mathcal{F}$
4:     Update the discriminator $F_d(f; \theta_d)$ by ascending the gradient
        $\nabla_{\theta_d} \left( \sum_{i=1}^{n} \log(1 - F_d(\hat{f}_i)) + \sum_{i=1}^{m} \log F_d(f_i) \right)$
5:     Update the classifier $F_c(f; \theta_c)$ by ascending the gradient ($a[i]$ indicates the $i$-th element of vector $a$)
        $\nabla_{\theta_c} \left( \sum_{i=1}^{m} \log F_c(f_i)[y_i^f] \right)$
6:     Update the generator $F_g(f; \theta_c)$ by ascending the gradient
        $\nabla_{\theta_g} \left( \sum_{i=1}^{n} \log F_c(F_g(F_e(v_i)))[y_i^v] + \sum_{i=1}^{m} \log F_d(F_g(F_e(v_i))) \right)$
7: **end while**

---

Once trained, $F_d(f; \theta_d)$ and $F_c(f; \theta_c)$ can be removed. Only $F_d(f; \theta_d)$ and $F_c(f; \theta_c)$ are used for face generation from voice in the inference. It is worth noting that the targeted scenario is *open-set*. In our evaluations, the model is required to work on previously unseen and unheard identities, *i.e.* $y^v \in \mathcal{Y}$ in the training phase, while $y^v \notin \mathcal{Y}$ in the testing phase.

## 4 Experiments

In our experiments, the voice recordings are from the `Voxceleb` [25] dataset and the face images are from the manually filtered version of `VGGFace` [26] dataset. Both datasets have identity labels. We use the intersection of the two datasets with the common identities, leading to 149,354 voice recordings and 139,572 frontal face images of 1,225 subjects. We follow the train/validation/test split in [25]. The details are shown in Table 1.

Separated data pre-processing pipelines are employed to audio segments and face images. For audio segments, we use a voice activity detector interface from the WebRTC project to isolate speech-bearing regions of the recordings. Subsequently, we extract 64-dimensional log mel-spectrograms using an analysis window of 25ms, with a hop of 10ms between frames. We perform mean and variance normalization of each mel-frequency bin. We randomly crop an audio clips around 3 to 8 seconds for training, but use the entire recording for testing. For the face data, facial landmarks in all images are detected using [5]. The cropped RGB face images of size $3 \times 64 \times 64$ are obtained by similarity transformation. Each pixel in the RGB images is normalized by subtracting 127.5 and then dividing by 127.5.

Table 1: Statistics of the datasets used in our experiments

|  | train | validation | test | total |
|---|---|---|---|---|
| # of speech segments | 113,322 | 14,182 | 21,850 | 149,354 |
| # of face images | 106,584 | 12,533 | 20,455 | 139572 |
| # of subjects | 924 | 112 | 189 | 1,225 |

**Training.** The network architecture is given in Table 2. The parameters in the convolutional layers of discriminator and classifier are shared in our experiments. We basically follow the hyperparameter setting in [29]. We used the Adam optimizer [14] with learning rate of 0.0002. $\beta_1$ and $\beta_2$ are 0.5 and 0.999, respectively. Minibatch size is 128. The training is completed at 100K iterations.

### 4.1 Qualitative Results

As a first experiment, we compared the outputs of the network in response to various noise signals to outputs obtained from actual speech recordings. Figure 2 shows outputs generated for four different types of noise. We evaluated noise segments of different durations (1,2,3, 5 and 10 seconds) to observe how the generated faces change with the duration. The generated images are seen to be

Table 2: The detailed CNNs architectures. For the voice embedding network, we use 1D convolutional layers. Conv $3_{/2,1}$ denotes 1D convoluitonal layer with kernel size of 3, where the stride and padding are 2 and 1, respectively. Each convolutional layer is followed by a Batch Normalization (BN) [9] layer and Rectified Linear Units (ReLU) [16]. The output shape is shown accordingly, where $t_{i+1} = \lceil (t_i - 1)/2 \rceil + 1$. The final outputs are pooled over time, yielding a 64-dimensional embedding. We use 2D deconvolutional layers with ReLU for the generator and 2D convolutional layers with Leaky ReLU (LReLU) for the discriminator and classifier. The final output is given by fully connected (FC) layer.

| Voice Embedding Network | | | Generator | | |
|---|---|---|---|---|---|
| Layer | Act. | Output shape | Layer | Act. | Output shape |
| Input | - | $64 \times t_0$ | Input | - | $64 \times 1 \times 1$ |
| Conv $3_{/2,1}$ | BN + ReLU | $256 \times t_1$ | Deconv $4 \times 4_{/1,0}$ | ReLU | $1024 \times 4 \times 4$ |
| Conv $3_{/2,1}$ | BN + ReLU | $384 \times t_2$ | Deconv $3 \times 3_{/2,1}$ | ReLU | $512 \times 8 \times 8$ |
| Conv $3_{/2,1}$ | BN + ReLU | $576 \times t_3$ | Deconv $3 \times 3_{/2,1}$ | ReLU | $256 \times 16 \times 16$ |
| Conv $3_{/2,1}$ | BN + ReLU | $864 \times t_4$ | Deconv $3 \times 3_{/2,1}$ | ReLU | $128 \times 32 \times 32$ |
| Conv $3_{/2,1}$ | BN + ReLU | $64 \times t_5$ | Deconv $3 \times 3_{/2,1}$ | ReLU | $64 \times 64 \times 64$ |
| AvePool $1 \times t_5$ | - | $64 \times 1$ | Deconv $1 \times 1_{/1,0}$ | - | $3 \times 64 \times 64$ |
| **Discriminator** | | | **Classifier** | | |
| Layer | Act. | Output shape | Layer | Act. | Output shape |
| Input | - | $3 \times 64 \times 64$ | Input | - | $3 \times 64 \times 64$ |
| Conv $1 \times 1_{/1,0}$ | LReLU | $32 \times 64 \times 64$ | Conv $1 \times 1_{/1,0}$ | LReLU | $32 \times 64 \times 64$ |
| Conv $3 \times 3_{/2,1}$ | LReLU | $64 \times 32 \times 32$ | Conv $3 \times 3_{/2,1}$ | LReLU | $64 \times 32 \times 32$ |
| Conv $3 \times 3_{/2,1}$ | LReLU | $128 \times 16 \times 16$ | Conv $3 \times 3_{/2,1}$ | LReLU | $128 \times 16 \times 16$ |
| Conv $3 \times 3_{/2,1}$ | LReLU | $256 \times 8 \times 8$ | Conv $3 \times 3_{/2,1}$ | LReLU | $256 \times 8 \times 8$ |
| Conv $3 \times 3_{/2,1}$ | LReLU | $512 \times 4 \times 4$ | Conv $3 \times 3_{/2,1}$ | LReLU | $512 \times 4 \times 4$ |
| Conv $4 \times 4_{/1,0}$ | LReLU | $64 \times 1 \times 1$ | Conv $4 \times 4_{/1,0}$ | LReLU | $64 \times 1 \times 1$ |
| FC $64 \times 1$ | Sigmoid | 1 | FC $64 \times k$ | Softmax | $k$ |

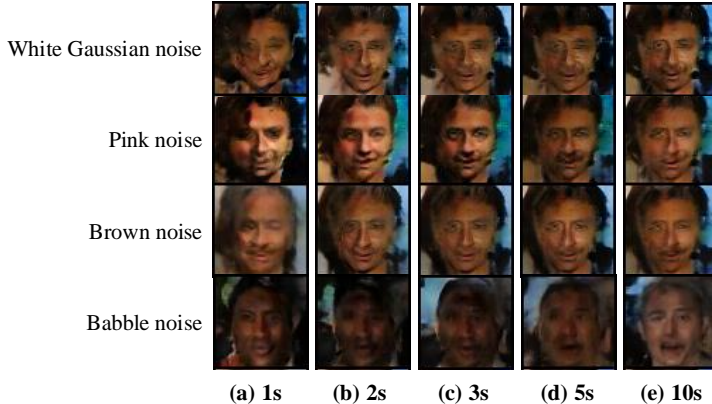

White Gaussian noise

Pink noise

Brown noise

Babble noise

**(a) 1s**  **(b) 2s**  **(c) 3s**  **(d) 5s**  **(e) 10s**

Figure 2: The generated face images from noise input. Each row shows the generated faces using one of the four noise audio segments with different durations. (a) 1 second, (b) 2 seconds, (c) 3 seconds, (d) 5 seconds, (e) 10 seconds.

blurry, unrecognizable and generally alike, since there is no identity information in noise. With longer noise recordings, the results do not improve. Similar results are obtained over a variety of noises.

On the other hand, when we use regular speech recordings with the aforementioned durations as inputs, outputs tend to be realistic faces, as seen in Figure 3. The results indicate that while the generator does learn to produce face-like images, actual faces are produced chiefly in response to actual voice. We infer that while the generator has learned to map the speech manifold to the manifold of faces, it maps other inputs outside of this manifold.

Figure 3 also enables us to subjectively evaluate the actual output of the network. These results are typical and not cherry-picked for presentation (several of our reconstructions match the actual speaker closely, but we have chosen not to selectively present those to avoid misrepresenting the actual performance of the system). The generated images are on the left, while the reference images (the actual faces of the speakers) are on the right. To reduce the perceptual bias and better illustration, we show multiple reference face images for each speaker. Although the generated and the reference

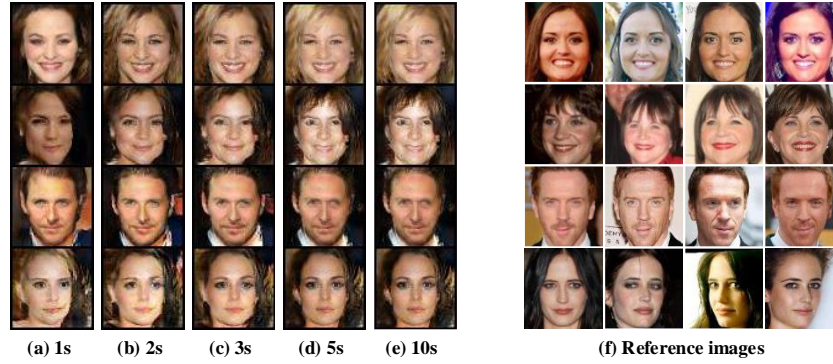

(a) 1s   (b) 2s   (c) 3s   (d) 5s   (e) 10s          (f) Reference images

Figure 3: (a)-(e) The generated face images from regular speech recordings with different durations. (f) the corresponding reference face images. These 4 speakers (from top to bottom) are Danica McKellar, Cindy Williams, Damian Lewis, and Eva Green.

face indicate different persons, the identity information of these two are matched in some sense (like gender, ethnicity, etc.). With longer speech segments, the generated faces gradually converge to faces associatable with the speaker. Figure 4 shows additional examples demonstrating that the synthesized images are generally age- and gender-matched with the speaker.

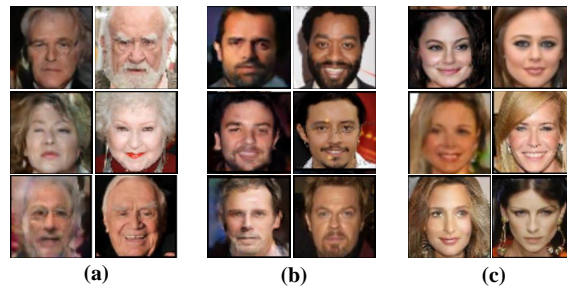

(a)                (b)                (c)

Figure 4: (a) Faces from old voices. (b) Faces from male voices. (c) Faces from female voices. For each group, images on the left are the generated images and images on the right are the references.

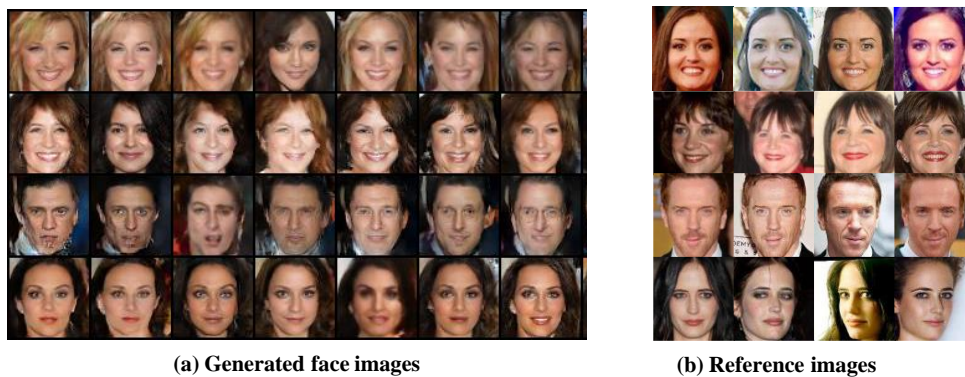

(a) Generated face images                (b) Reference images

Figure 5: The generated face images from different speech segments of the same speaker and their corresponding face images. Each row shows the results from the same speaker.

In the next experiment, we select 7 different speech segments of each speaker and generate the faces. Entire recordings are used. The results are shown in Figure 5 (reference images are also provided for comparison). Once again, the results are typical and not cherrypicked. We believe that the images in each row exhibit reasonable variations of the same person (except the fourth image in the first row), indicating that our model is able to build a mapping between the speech group to face group, thus retaining the identity of the face across speech segments from the same speaker.

## 4.2 Quantitative Results

We attempt to quantitatively distinguish between the faces generated in response to noise from those generated from voice using the discriminator $F_d()$ itself. Note that $F_d()$ is biased, and has explicitly been trained to tag synthesized faces as fake. The mean and standard deviation (obtained from 1000 samples) of its output value in response to actual voices are 0.09 and 0.13 respectively, while for (an identical number of) noise inputs they are 0.06 and 0.07 respectively. Even a biased discriminator is clearly able to distinguish between faces generated from voices, and those obtained from noise.

As a first test, we attempted to ID (classify) faces derived from test recordings of the speakers in the training set. On a set of 4620 recordings, from 924 identities, we achieved a top-1 accuracy of 61.7% and a top-5 accuracy of 82.3%, showing that the voice-based face reconstructions *do* actually faithfully capture the identities of *known* subjects.

For unknown subjects, we ran a gender classifier on 21,850 generated faces (from the speech segments in test set). The classifier was trained on the face images in the training set using the network architecture of the discriminator, with an accuracy of 98.97% on real face images. The classifier obtained a 96.45% accuracy in matching the gender of the generated faces to the known gender of the speaker, showing that the generated faces are almost always of the correct gender.

Finally, we evaluated our model by leveraging the task of voice to face matching. Here, we are given a voice recording, a face image of the true speaker, and a face image of an imposter. We must match the voice to the face of the true speaker. Ideally, the probe voice could be replaced by the generated face image if they carry the same identity information. So the voice to face matching problem reduces to a typical face verification or face recognition problem. The resulting matching accuracy could be used to quantitatively evaluate the association between the speech segment and the generated face.

We construct the testing instances (a probe voice recording, a true face image, and an "imposter" face) using data in the testing set, leading to 2,353,560 trials. We also compute the matching accuracy on about 50k trials constructed from a small part of the training set to see how well the model fit to the training data. We also perform stratified experiments based on gender where we select the imposter face with the same gender as the true face. In this case, gender information cannot be used for matching anymore, leading to a more fair test.

The results are shown in Table 3. The high accuracies obtained on the training set for the unstratified and gender stratified tests (96.83% and 93.98% respectively) show that generated faces do carry correct identity information for the training set. The results on the test set on unstratified and gender stratified tests (76.07% and 59.69% respectively) are better than those in DIMNets-G [39], indicating that our model learns more associations than gender. The large drop compared to the results on the training set shows however that considerable room remains to improve generalizability in the model.

Table 3: The voice to face matching accuracies. Our results are given by replacing the probe voice embeddings by the embeddings of the generated face.

|  | unstratified group (ACC. %) (training set / testing set) | stratified group by gender (ACC. %) (training set / testing set) |
|---|---|---|
| SVHF [25] | - / 81.00 | - / 65.20 |
| DIMNets-I [39] | - / 83.45 | - / 70.91 |
| DIMNets-G [39] | - / 72.90 | - / 50.32 |
| ours | 96.83 / 76.07 | 93.98 / 59.69 |

## 5  Discussion and Conclusion

The proposed GAN-based framework is seen to achieve reasonable reconstruction results. The generated faces have identity associations with the true speaker. There remains considerable room for improvement. Firstly, there are obvious issues with the GAN-based output: the produced faces have features such as hair that are presumably not predicted by voice, but simply obtained from their co-occurrence with other features. The model may be more appropriately learned through data cleaning that removes obviously unrelated aspects of the facial image, such as hair and background. The proposed model is vanilla in many ways. For instance [34, 36] describe several explicit correspondences between speech and face features, e.g. different phonetic units are known to relate to different facial features. We are investigating models that explicitly consider these issues.

## Footnotes

[1] A version of the work reported in this paper was tested live by nearly 1000 people at the World Economic Forum in Tianjin, Sep 2018.

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
