[Reviews · NeurIPS 2019]

Reviewer 1



The paper suggests to do something that strains belief -- create an estimate of a face from a speech signal. Surprisingly, this is able to succeed in some regards! Ultimately this stems from the fact that speech production relies on face characteristics for aspects of the sound, a fact that has been known in that literature for a while. The paper is thought-provoking and leads one to wonder what other reconstruction tasks might be possible across domains that seem unlinked. The quantitative evaluation is a little tricky to do correctly. It would be interesting to see if there is a way to more directly compare, for example, see whether people are able to pick out which of the GAN faces was created by the model (given the human face), or have people choose which face is most likely to have created the speech signal they hear. This might support whether people have the same judgements as the GAN model.

Reviewer 2



Detailed comments already given above.

Reviewer 3



This paper proposes a convolutional neural network based model to reconstruct a face from spoken speech. The training is done by using supervised GAN. The problem is novel, but the model itself is not so much as using encoder (or embedder) and decoder (or generator) is quite standard, and supervised GAN training has also been popularly used, so in that perspective, its novelty is incremental. But I think this paper needs more thorough experimental study to show the effectiveness of the proposed model: 1. From the experimental results, I suspect that the generated faces only match those attributes (gender, race, etc.) but not much about identities. I propose face identity classification might be better in this regard rather than the illustrated matching test. Even the matching test results with test data and the same gender, the accuracy was slightly higher than chance so it does not seem significant. 2. There is no baseline. In qualitative results, there was no baseline result. I understand that the problem might be new, but at least the paper might present ablation study or simpler baselines. The models in Table 3 were without any explanation and I assume they are only for matching test, not for generative models for speech to face reconstruction task. In sum, I think the problem looks interesting and novel, but the novelty of proposed model seems incremental, and the experimental results do not look to demonstrate the effectiveness of the proposed model enough. Writing looks fine but can be improved. Typos: line #173: Only F_d() and F_c() are used -> Only F_e() and F_g() are used line #197: a first -> the first

[Author Response · NeurIPS 2019]

We sincerely thank all the reviewers for their helpful comments. We attempt to address their concerns below.

## Additional evaluations and human judgment

● All reviewers recommend additional evaluations. ● Reviewer 1 also suggests human evaluator opinion tests. ● Reviewer 2 asks if humans are able to visualize faces for voices. ● Reviewers 2 and 3 wonder if the reconstructed faces do indeed capture the speaker's ID. ● Reviewer 3 recommends comparison to baselines. While we are unable to compare to existing baselines, since there is no prior work on the topic, we have run additional experiments which we hope address the rest of the reviewers' concerns under this topic. These results will be included in the paper and the experimental setups and data made public.

*a. Do the generated faces actually capture the speaker's ID*: The GAN is, in fact, optimized for face identification – a critical component of the loss is provided by a face-ID system (which is separately trained, and not optimized with the GAN). To explicitly address the question, we ran the face-ID system on a set of faces derived from novel speech recordings that are not part of the training set. The test is a "closed-set" test, in the sense that the IDs are expected to be one of the 924 IDs represented in the ID system. Table 1 shows the top-1 and top-5 identification results. While not perfect, the results clearly show that the faces generated from the voices clearly do capture the ID of the speaker.

Table 1: Results of closed-set recognition

| Total number of samples | # of samples per class | # of classes | Top-1 accuracy | Top-5 accuracy |
|---|---|---|---|---|
| 4620 | 5 | 924 | 61.7% | 82.3% |

*b. Human judgment test*: Studies in experimental psychology have shown that humans *are* indeed significantly better than chance at matching voices to faces, e.g. [1].

We ran a human judgment test, as suggested by the reviewer. A total of 20 volunteers (who had no prior knowledge of this work) were each presented 20 trials. Each trial comprised a voice and two GAN faces, one of which was produced from the voice. The subject was required to identify the generated face for the voice. To eliminate extraneous factors, we tried to match covariates, e.g. both faces were always the same gender. Trial voices and faces were not repeated for any subject. Table 2 shows the results. The results are much better than chance, with a P value below 0.001, and are similar to performances reported in human studies, e.g. [1].

Table 2: Human subject accuracy in matching voices to generated faces

| Minimum accuracy | Maximum accuracy | Mean accuracy | Standard deviation |
|---|---|---|---|
| 55.0% | 75.0% | 67.5% | 8.1 |

[1] Mavica, L. W., & Barenholtz, E. (2013). Matching voice and face identity from static images. Journal of Experimental Psychology: Human Perception and Performance, 39(2), 307-312

## Why GAN architectures?

We attempted a variety of regression and GAN architectures. The GAN architecture that included a Face-ID loss turned out to be the most effective. The key component was the face-ID system that provides a loss to optimize the generator.

## Predicting age from voice, which was not shown with high accuracy (reviewer 2)

Age prediction from voice is a well-studied problem, with mean absolute errors of less than 5 years being achievable by systems trained with age-labeled voice data.

In our setup, the challenge was the lack of appropriate training data, as the reviewer rightly points out. Voxceleb, has many voice recordings and face images for each person, however there is no information about the correspondence in age between them. Thus, when we pair a voice with a face during training, the voice may have been recorded when the subject was young, while the facial image may have been taken when they were much older. As a result, there really is no assurance of learning age associations, although remarkably, some association seems to be learned, presumably due to some degree of correspondence in the training data.

## What would be an application for such a task? (reviewer 2)

We have had expressions of interest from the entertainment and gaming industries (e.g. to generate avatars). The research is directly funded by law enforcement agencies in the US, although they only expect to use some carefully-vetted aspects of it. The research itself was not, however, targeted at a specific application, but to determine if visual information beyond the obvious covariates could somehow be extracted from voice. The answer seems to be affirmative.

[Meta-Review · NeurIPS 2019]

The paper proposes a very novel method that creates an estimate of a face from a voice and works as a supervised method . The reviewers initially were not so convinced and with some disagree. They requested additional information. The rebuttal was satisfying so that also one reviewer changed its score from weak rejection to acceptance. Thus, after a discussion with the Senior Area chair, the paper is accepted . This meta-review was reviewed and revised by the Program Chairs, based on discussions with the Senior Area Chair.